# Quantifying White Matter Hyperintensities: Automated Volumetry Compared with Visual Grading Scales

**DOI:** 10.3390/medicina62010060

**Published:** 2025-12-28

**Authors:** Arturs Titovs, Artūrs Šilovs, Kalvis Kaļva, Ardis Platkājis, Andrejs Kostiks, Kristīne Šneidere, Guntis Karelis, Ainārs Stepens, Nauris Zdanovskis

**Affiliations:** 1Department of Radiology, Riga Stradins University, LV-1007 Riga, Latvia; 048552@rsu.edu.lv (A.T.); arturs.silovs@rsu.lv (A.Š.); kalvis.kalva@rsu.lv (K.K.);; 2Department of Radiology, Riga East University Hospital, Hipokrata iela 2, LV-1038 Riga, Latvia; 3Institute of Public Health, Riga Stradins University, Dzirciema iela 16, LV-1007 Riga, Latviaainars.stepens@rsu.lv (A.S.); 4Department of Neurology and Neurosurgery, Riga East University Hospital, Hipokrata iela 2, LV-1038 Riga, Latvia

**Keywords:** white matter hyperintensities, cognitive impairment, automated volumetric quantification, neuroradiology, Fazekas scale, Montreal Cognitive Assessment, magnetic resonance imaging, regional white matter hyperintensities, total white matter hyperintensities, white matter hyperintensity volume, quantitative neuroimaging biomarkers

## Abstract

*Background and objectives.* White matter hyperintensities (WMHs) on brain magnetic resonance imaging (MRI) are linked to cognitive decline, but clinical assessment still relies mainly on visual grading (Fazekas), which is coarse and rater-dependent. We described the lesion volume of WMHs and the association of the anatomical distribution with the severity of cognitive impairment using automated lesion analysis. In addition, we evaluated whether automated volumetric quantification is more strongly associated with cognitive performance than visual grading. *Materials and Methods.* In a retrospective cross-sectional study, forty-one adults referred for cognitive concerns underwent standardised 3.0 tesla MRI. White matter hyperintensities were automatically segmented using Icometrix software to obtain total and regional volumes (periventricular, subcortical, brainstem, cerebellum). Visual grading used the Fazekas scale separately for periventricular and deep white matter, with a combined grade defined by the higher of the two. Cognitive performance was grouped based on the Montreal Cognitive Assessment (MoCA) into high (≥26), moderate (18–25), and low (≤17). Statistics included Spearman’s correlation and the Kruskal–Wallis test with Dunn’s post hoc test where applicable. *Results.* Higher total white matter hyperintensity volume was associated with lower Montreal Cognitive Assessment scores and showed significant differences across cognitive groups. The Fazekas combined grade correlated more weakly with the MoCA score. Regional volumetric differences showed trends, but were not statistically significant. Total volumetric burden increased stepwise across combined Fazekas categories, supporting convergent validity between methods. *Conclusions.* Our study found that automated volumetric quantification provides a more objective, sensitive, and scalable measure of white matter hyperintensity burden than visual grading, aligns more closely with cognitive status, and is better suited for longitudinal monitoring and research endpoints.

## 1. Introduction

White matter hyperintensities (WMHs) are a common finding on magnetic resonance imaging (MRI) of the brain, and for many years they have been associated with cognitive impairment and various neurological and cognitive disorders [1,2,3]. Even nowadays, the pathophysiological mechanism of these white matter changes is not completely clear and appears to be multifactorial; however, it is often considered a consequence of cerebral small-vessel disease (cSVD), primarily associated with elevated blood pressure, hyperlipidaemia, and diabetes [3,4]. Moreover, factors such as chronic cerebral hypoperfusion have been observed to be involved in the development of white matter lesions and, in turn, increasing cerebral amyloid-beta (Aβ) burden in patients with impaired cognition [4,5,6,7,8]. Animal studies have demonstrated the connection between cerebral hypoperfusion and white matter damage, as well as an increase in cerebral amyloid-beta burden, with the formation of white matter lesions, further implicating the role of WMH in cognitive impairment [6,7]. Microglial, mural cell, astrocytic, and oligodendrocytic dysfunctions contribute to the formation of WMH, as well [9]. Besides these factors, endothelial cell inflammation has also been associated with white matter degeneration [10].

One study estimates that roughly 15% of adults aged 50 years or more are affected by mild cognitive impairment (MCI). Moreover, it supports timely assessment and treatment of MCI in adult risk groups as a measure of prevention of dementia [11]. It has been reported that the presence and severity of white matter changes can be a predictive factor in the development of dementia in patients with MCI. In addition, the rate at which WMH progresses influences the speed at which mental processing declines [6,12]. This highlights the link between cognitive disorders and WMH.

Due to projected trends in population ageing and growth, the number of individuals affected by dementia is expected to rise and almost triple by the year 2050, reaching roughly 150 million cases in the world. Besides the prevalence of dementia cases, the socioeconomic burden caused by dementia is projected to increase, as well. It is noted that dementia contributes to burnout in healthcare workers and increases fatigue and emotional distress among caregivers and family members. In addition, young-onset dementia is presumed to exacerbate the aforementioned adverse effects. Therefore, it is crucial to investigate and understand the relationship between white matter changes and cognitive function to prematurely detect and potentially avoid early cognitive decline [13,14,15,16].

White matter hyperintensities are mainly evaluated in two distinct ways: (1) visual rating scales and (2) fully automated quantitative analytic processing streams. Despite advancements in the field of automated segmentation methods, visual rating scales, such as the Fazekas scale, remain the most commonly used method to study WMHs [17,18]. The Fazekas scale was the first to rate periventricular white matter lesions (PVWML) and deep white matter lesions (DWML) independently. There are etiological, histopathological, and clinical differences between these types of lesions. PVWML are associated with cognitive impairment and increase the risk of developing dementia. DWM lesions are linked to mood disorders and their consequences, as well as motor disorders. Moreover, in already depressed patients, the level of cognitive impairment was predominantly associated with DWML instead of PVWML [17,18,19,20]. However, a systematic review performed by Bolandzadeh et al., shows that in some studies a significant link between decreased cognitive functions and DWM damage was observed [21].

The use of visual rating scales is helpful because of the human ability to observe and interpret patterns of WMH and isolate them from artefacts. Moreover, visual rating scales are essentially the only tools available to radiologists without access to automated quantitative techniques. However, visual rating scales are time-consuming, are notably susceptible to inter-rater and intra-rater variability, and suffer from low reliability, especially if the raters lack training. Taking these factors into account, these scales are not realistic or optimal for large-scale studies. In addition, visual rating scales are not able to provide information about regional WMH volumes [17,18].

Advances in medical imaging, computational power, and broader access to neuroimaging datasets have enabled the development of machine learning approaches for automated detection, quantification, and classification of white matter hyperintensities. The essence of machine learning is the use of sequential steps in order to employ computer algorithms to pinpoint clinically relevant areas and features [22]. White matter changes are often quantified and their spatial distribution analysed to better understand their relationship with clinical symptoms. In addition, the volumetric analysis of WMH has become more prevalent due to advances in and the availability of fully automated techniques [23]. The development of automated algorithms for detecting and quantifying white matter hyperintensities has been an active area of research aimed at improving the accuracy and reliability of these measurements. Accurate quantification and localisation of these white matter hyperintensities could provide valuable insights into the underlying mechanisms of cognitive decline [23,24].

The study aims to determine the volume of WMHs and to explore the association between the anatomical distribution of WMHs and the severity of cognitive impairment using automated quantitative lesion analysis. Another aim is to evaluate whether quantitative analysis is more strongly associated with cognitive impairment than a visual grading scale. Our study aims to provide insights into this relationship. The clinical need arises from the limitations of the Fazekas scale as a subjective, coarse, and rater-dependent tool. Automated volumetric WMH quantification may offer a more sensitive and reproducible biomarker of cognitive impairment, yet few studies have directly compared these approaches within the same clinical cohort. The existing literature provides limited insight into the association between quantitative WMH burden and cognitive impairment; this study aims to address this gap.

## 2. Materials and Methods

We conducted a retrospective cross-sectional analysis of individuals who underwent cognitive testing followed by brain MRI scans.

A total of 41 participants were included in our study and stratified into three cognitive groups according to their Montreal Cognitive Assessment (MoCA) scores [25,26,27]:High cognitive performance (HP) group (participants with MoCA scores ≥ 26);Moderate cognitive performance (MP) group (participants with MoCA scores ≥ 18 and ≤25);Low cognitive performance (LP) group (participants with MoCA scores ≤ 17).

There were 9 participants in the HP group (mean age 65, SEM 3.8, SD 11.5, youngest participant 44 years old, oldest 77 years old, mean MoCA score 28.2, SEM 0.4, SD 1.1, lowest score 27, highest score 30).

There were 18 participants in the MP group (mean age 70.3, SEM 1.7, SD 7.4, youngest participant 57 years old, oldest 81 years old, mean MoCA score 22.7, SEM 0.5, SD 2.3, lowest score 18, highest score 25).

There were 14 participants in the LP group (mean age 75.9, SEM 3.0, SD 11.1, youngest participant 62 years old, oldest 96 years old, mean MoCA score 9.7, SEM 1.2, SD 4.4, lowest score 4, highest score 16).

All continuous variables were tested for normality, and after the Shapiro–Wilk test, with *p* < 0.05 rejecting normality, the only exception was observed in the “Age” variable, in which data were distributed normally (see Table 1).

Demographic data, gender, and MoCA scores for participants are shown in Table 2.

Pearson’s Chi-Square test on gender was conducted, and there were no statistically significant differences between the groups (χ^2^ = 2.296, *p* < 0.317).

A Kruskal–Wallis H test was performed to evaluate age differences between groups, and no statistically significant differences were found between the groups (χ^2^ = 4.702, *p* < 0.095).

A Kruskal–Wallis H test was performed to examine differences in MoCA scores between groups, and statistically significant differences were found (χ^2^ = 34.802, *p* < 0.001).

### 2.1. Selection of Participants

Participants enrolled in this study were individuals referred to a neurologist because of self-reported cognitive difficulties or concerns about cognitive decline raised by their primary care physician.

A board-certified neurologist with established clinical expertise in evaluating and treating cognitive impairment oversaw the diagnostic assessments in this study.

The study excluded participants with clinically relevant neurological disorders or psychiatric conditions (such as vascular malformations, a history of tumours, intracerebral lobar haemorrhages, severe strokes, Parkinson’s disease, major depressive disorder, schizophrenia, manic episodes, bipolar disorder, etc.), as well as any history of substance abuse.

Besides WMHs, no other clinically significant pathologies were observed on the MRI scans of patients enrolled in this study. None of the participants displayed signs of cerebral amyloid angiopathy or any other neurodegenerative diseases. No participant had vascular malformations, more than four microbleeds, intra- or extra-axial tumours. Based on available clinical records, none of the participants had uncontrolled hypertension, other vascular diseases, or diabetes. All participants had completed at least 16 years of education and held university degrees.

### 2.2. PVWM and DWM Hyperintensity Grading Scale

We utilised Fazekas scale grading by labelling patient T2 FLAIR (fluid-attenuated inversion recovery) images ranging from Grade 1 to Grade 3, and we assigned separate grades according to lesion location in periventricular white matter or in deep white matter [28].

PVWMH Grade 1 was assigned to patients with mild hyperintensities that were found along the ventricular horns, also known as pencil-thin lining or “caps.” Grade 2 was assigned to patients exhibiting more hyperintensities, and Grade 3 was assigned to patients with lesions that extended into DWM (see Figure 1).

DWMH Grade 1 was assigned when small, punctate foci in subcortical regions were observed, Grade 2 was assigned to patients with a greater number of foci along with the early formation of white matter hyperintensity confluences, and Grade 3 was assigned to patients with large confluent areas of white matter hyperintensities (see Figure 2).

A Fazekas scale combined grade was assigned based on the highest grade between two lesion locations. For example, if a patient had a PVWMH Grade 1 and DWMH Grade 2, the combined grade would be 2.

### 2.3. MRI Acquisition Protocol and White Matter Hyperintensity Automated Quantification and Delineation

We utilised a 3.0 Tesla General Electric (GE) MRI scanner in a university hospital to perform MRI scans on all study participants. The following sequences were used:3D T1 axial (technical parameters—flip angle 11, TE (time to echo) min full, TI (inversion time) 400, FOV (field of view) 25.6, layer thickness 1 mm);3D FLAIR sagittal (technical parameters—TE 119, TR (repetition time) 4800, TI 1473, echo 182, FOV 25.6, layer thickness 1.2 mm);High-resolution hippocampal structure assessment sequence (technical parameters—flip angle 122, TE 50, Echo 1, TR 8020, FOV 17.5, layer thickness 2, coronal direction perpendicular to the hippocampus);DWI (technical parameters—b = 0, 1000, and synthetic 2000 s/mm^2^, flip angle 90, TE 76.0, TR 9852.0, slice thickness 3 mm);SWI (technical parameters—flip angle 15, TE 22.5, TR 34.7, slice thickness 3 mm).

In addition, 3D FLAIR T2 (TR 6002.0, TE 136.6, slice thickness 2 mm) scans were performed in order to employ the Fazekas scale to assign a grade.

WMHs were quantified and delineated using Icometrix software (icobrain tbi report for MRI, version 5.12.0; Leuven, Belgium). The software identifies and evaluates WMHs by total volume and further categorises the lesions based on their anatomical regions (subcortical, periventricular, brainstem, and cerebellum), displaying volume values for each region. The software evaluates the volume of the whole brain, cortical grey matter, and the hippocampus. Moreover, Icometrix software compares these values to age- and sex-normative references from population data; however, in this study, we focused on WMHs (see Figure 3) [29].

The Icometrix method employs sequential steps to segment and quantify WMH lesions. The first step requires 3D T1 and 3D FLAIR images as inputs. Afterwards, the 3D FLAIR image is co-registered to the 3D T1 image for spatial alignment. The next step is pre-processing, which involves skull stripping, followed by non-rigid atlas-to-image registration and bias field correction, followed by brain tissue segmentation based on GM (grey matter), WM (white matter), and CSF (cerebrospinal fluid). The subsequent step is lesion segmentation, which involves lesion detection, lesion filling, and lesion refinement or “pruning”, during which artefacts or partial-volume effects are removed. Ultimately, volume calculation of the whole brain, WM, and GM is performed, and WMH lesions are quantified and delineated [30,31,32].

### 2.4. Statistical Analysis

JASP (version 0.19.3; Amsterdam, The Netherlands) was used for statistical analysis [33]. Descriptive statistics were used to summarise the data, the Shapiro–Wilk test to evaluate data normality, and non-parametric tests for data analysis based on the results of the Shapiro–Wilk test. Additionally, Pearson’s Chi-Square test, the Kruskal–Wallis test, the Kruskal–Wallis H test, and Dunn’s post hoc analysis were performed. Spearman’s correlation test was performed to determine the relationship between MoCA test results and the automatically measured volume of WMH, as well as the assigned Fazekas scale grade to PVWMH and DWMH and the combined grade. A Kruskal–Wallis test was performed to establish statistically significant differences between the 3 groups based on the MoCA score and WMH volumes, as well as the assigned Fazekas scale grade to PVWMH and DWMH, and the combined grade. When statistically significant results were observed, post hoc pairwise comparisons were performed using Dunn’s test, with Bonferroni and Holm adjustments.

Outliers have not been removed; however, they can be observed by noting the confidence interval that is shown on every raincloud plot.

## 3. Results

### 3.1. Fazekas Scale Grade (Visual Rating Scale) Correlation and Statistical Significance Analysis with MoCA

Spearman’s correlation was used to assess the relationship between Fazekas scale grades assigned to PVWM and DWM, the combined grade, and the MoCA score. The results showed statistically significant negative correlations only between the Fazekas scale combined grade and the MoCA score (PVWM grade (r = −0.286, *p* = 0.070), DWM grade (r = −0.235, *p* = 0.139), combined grade (r = −0.325, *p* = 0.038) (see Table 3)).

Kruskal–Wallis tests were performed to observe statistical significance between Fazekas scale grades and MoCA groups.

By performing the Kruskal–Wallis test, statistical significance was found between the PVWM grade and MoCA groups (H (2) = 6.991, *p* < 0.030) (see Figure 4).

Based on the statistically significant Kruskal–Wallis test, Dunn’s post hoc test was conducted, finding statistically significant differences between the MP and LP groups (*p* < 0.009; after Bonferroni and Holm correction, statistical significance was maintained) (see Table 4).

The Kruskal–Wallis test did not find statistically significant differences between the DWM grade and MoCA groups (H (2) = 2.859, *p* < 0.239) (see Figure 5).

The Kruskal–Wallis test showed statistical significance between the combined grade and MoCA groups (H (2) = 7.151, *p* < 0.028) (see Figure 6).

After performing Dunn’s post hoc test, statistically significant differences were found between the MP and LP groups (*p* < 0.008; after Bonferroni and Holm correction, statistical significance was maintained) (see Table 5).

### 3.2. White Matter Hyperintensities

The mean total volume of WMHs in our study was 6.74 mL.

The highest mean volume load of WMHs was observed in the periventricular region with a mean value of 3.525 mL, followed by 3.161 mL in the subcortical region, 0.046 mL in the brainstem, and 0.002 mL in the cerebellum.

Between MoCA groups, the highest mean total volume of WMHs was observed in the LP group, with a mean of 11.049 mL. This observation holds when we evaluate the mean volume in the LP group for each region separately. The periventricular region in the LP group had the highest measured mean WMH volume of 6.466 mL, followed by a subcortical mean WMH volume of 4.521 mL and a brainstem mean WMH volume of 0.057 mL, also found in the LP group. And, finally, the cerebellum mean WMH volume was the same between the LP and MP groups at 0.003 mL.

The lowest mean total volume load of WMHs was observed in the MP group, with a mean total volume of 4.348 mL; the same was observed in the periventricular region, where the lowest mean volume load of WMHs was also found in the MP group (1.821 mL). In the remaining regions, the lowest mean WMH volume load was observed in participants in the HP group (see Table 6 and Table 7).

Out of 41 participants enrolled in the study, only 1 participant (2.4%) had WMHs only in a single region (subcortical), while 28 participants (68.3%) had WMHs in two regions, 10 participants (24.4%) had WMHs in three regions, and 2 participants (4.8%) had WMHs in four regions.

A Spearman’s correlation test was performed to determine the relationship between lesion load and the overall MoCA score. The results showed a statistically significant negative correlation only between total WMH volume (r = −0.392, *p* = 0.011) and the MoCA score. This implies that a higher total WMH load correlates with a lower MoCA score (see Table 8).

Kruskal–Wallis tests were performed to observe statistical significance between total and regional WMH volumes and MoCA groups.

By performing the Kruskal–Wallis test, no statistical significance was found between subcortical WMH volume and MoCA groups (H (2) = 3.079, *p* < 0.215) (see Figure 7).

Upon performing the Kruskal–Wallis test, statistically significant differences were not found between periventricular WMH volume and MoCA groups (H (2) = 5.206, *p* < 0.074) (see Figure 8).

The Kruskal–Wallis test did not detect statistical significance between cerebellum WMH volume and MoCA groups (H (2) = 0.604, *p* < 0.739) (see Figure 9).

The Kruskal–Wallis test did not detect statistical significance between brainstem WMH volume and MoCA groups (H (2) = 0.964, *p* < 0.617) (see Figure 10).

By performing the Kruskal–Wallis test, statistically significant differences were found between total WMH volume and MoCA groups (H (2) = 6.998, *p* < 0.030) (see Figure 11).

By conducting Dunn’s post hoc test, statistically significant differences were found between the HP and LP groups (*p* < 0.023; after Bonferroni and Holm correction, statistical significance was not maintained) and between the MP and LP groups (*p* < 0.024, but after Bonferroni and Holm correction, there were no statistically significant differences) (see Table 9).

### 3.3. Comparison of Total White Matter Hyperintensity Volume and Fazekas Scale Combined Grade

The Kruskal–Wallis test was performed to observe statistical significance between total WMH volumes and the combined grade. Statistically significant differences were found between total WMH volume and combined grade groups (H (2) = 29.461, *p* < 0.001) (see Figure 12).

After performing Dunn’s post hoc test, statistically significant differences were found between the 1 and 2 groups (*p* < 0.001; after Bonferroni and Holm correction, statistical significance was maintained), between the 1 and 3 groups (*p* < 0.001; after Bonferroni and Holm correction, there were statistically significant differences), and between the 2 and 3 groups (*p* < 0.008; after Bonferroni and Holm correction, statistical significance was maintained) (see Table 10).

## 4. Discussion

The global trend of population ageing and the growing number of patients with various cognitive impairments underscore the importance of identifying biomarkers associated with cognitive decline. WMHs have been established as a significant biomarker for cognitive deterioration.

We analysed hyperintensities of PVWM and DWM and the combined grade and observed a correlation only between the combined grade and the MoCA score. The analysis of hyperintensities of PVWM and DWM and the combined grade displayed statistically significant differences in PVWM and combined grade between MP and LP groups in both cases. No statistically significant differences in DWM hyperintensities were found between the groups. These results support the findings of several studies. However, they conflict with the findings of one specific study, which claimed that DWMH volume inversely correlates with cognitive performance [19,21,34,35].

In our study, the mean total volume of WMHs was 6.74 mL. We observed the highest mean volume load of WMHs in the periventricular region. The participants in the LP group had the highest mean total WMH volume—11.049 mL. However, the lowest mean WMH volume load was observed in the MP group, at 4.348 mL. The observed variability in WMH volumes provides further evidence of a strong connection between white matter integrity and cognitive functioning.

There is a limited number of studies that quantify the volume of WMH and correlate the values to cognitive performance; however, in one meta-analysis, the mean total volume of WMHs was in a significantly broad range, from 2 to 9 mL. Perhaps this finding can be explained by a plethora of factors that differ in each study, such as age, comorbidities, and methods of WMH quantification [36]. Nevertheless, a common finding among many studies is the prevalence of a larger volume of PVWMH compared to the volume of DWMH [37]. The same pattern was noted in our study, as well.

Most participants in our study had WMH lesions in two distinct locations, followed by three locations. Only a few participants had WMH lesions in four and one location. These findings indicate that WMH lesions primarily affect multiple white matter areas and are extremely rarely confined to a single location. Previous research also shows involvement of various white matter areas. However, the definition of which parts of the white matter belong to a particular area is highly variable across studies, which could be contributing to inconclusive and varied results and may be making comparisons difficult. Keeping in mind this limitation, newer studies suggest using voxel-level probability mapping or “bullseye” parcellation [20,23,38,39]. These methods are further supported by growing evidence that specific cognitive disorders stem from damage to certain areas of white matter [40].

Our results displayed a negative correlation only between total WMH volume and the MoCA groups. Statistically significant differences were observed between total WMH volume and the HP-LP group and between MP and LP groups; however, after Bonferroni and Holm correction, statistical significance was not maintained in either case. This finding is in line with a large number of studies showing that higher WMH burden is associated with lower cognitive performance [41]. Only a single study exclusively evaluated the total WMH volume and its relationship to specific cognitive functions, and it only found an association between increased total WMH volumes and lower levels of perceptual speed [42].

We did not find statistically significant differences between periventricular WMH, subcortical WMH, brainstem WMH, and cerebellum WMH and the participants’ groups. This finding can be explained by a plethora of possible factors. Firstly, the nature of the MoCA test captures overall cognitive status rather than domain-specific functions. Regional WMH volumes represent localised lesion distributions that may not independently have a measurable effect on a global cognitive screening test, such as the MoCA test. Secondly, variability in WMH distribution across participants may dilute regional effects on cognitive performance. Patients with a similar total WMH burden are likely to have WMH lesions in vastly different regions; therefore, upon analysing regional WMH volumes, this factor is expected to dilute region-specific effects, as patients with minimal involvement in a specific region are grouped with those that display a significant regional burden. This may mask the actual regional contribution to cognitive impairment. From a mechanistic perspective, cognitive impairment related to WMH is believed to arise from distributed network disconnection rather than focal damage to a single anatomical region. Small to moderate lesion loads across multiple white matter regions tend to create a cumulative burden effect, impairing cognitive performance, an effect more accurately highlighted by total WMH volume rather than by regional lesion volume alone. Typically, periventricular regions are heavily affected by WMH lesions and are associated with cognitive decline, but infratentorial regions are rarely involved [39,43]. The latter was observed in our study, as well.

We observed statistically significant differences in total WMH volume between the combined grade groups, specifically between the 1–2 and 1–3 groups, which remained significant after correction. This finding is supported by a few studies that show that WMH volumes and Fazekas scale grades are highly correlated. In addition, plenty of studies mention that quantitative volumetric analysis is more accurate, objective, and more effective at assessing WMHs compared to visual scales, such as the Fazekas scale [44,45].

Moreover, volumetric analysis can detect unusual WMH patterns and classify advanced WMH lesions [39,46].

Several studies demonstrate that WMH volumetric analysis is superior to visual scales, such as the Fazekas scale, in correlating WMHs to cognitive impairment [47,48].

A practical advantage of WMH quantitative analysis methods is the ability to use data in longitudinal studies and correlate changes in WMH volumes to clinical outcomes and the level of cognitive impairment. A few studies observed the longitudinal changes of WMHs and concluded that the increase in WMHs leads to worse cognition, increases the chance of progression to dementia, and leads to poorer functional and clinical outcomes; however, regression of WMH volumes improves clinical outcomes. Based on this, it is essential to discover nonpharmacological and pharmacological interventions that decrease the burden of WMHs. Additionally, monitoring changes in WMH volume is possible by utilising quantitative analysis tools [49,50,51,52].

In a study which analysed multiple ischaemic stroke cohorts, it was observed that a higher baseline of WMH volume was associated with worse poststroke cognitive functioning, independent of acute stroke volume or old infarcts, indicating that greater WMH burden is a predictor of worse cognitive outcomes among patients affected by stroke [51]. In addition, a study of poststroke patients found that following physical activity guidelines, the volume of WMHs decreased compared to the baseline [53]. Several studies have established that the use of antihypertension medications causes a decrease in WMH volume [49,54].

Modern automated WMH pipelines are now independently validated with low false-positive rates and regional/global fidelity (TrUE-Net), accompanied by consensus best-practice guidance for deployment, and they already predict outcomes in vascular cohorts, supporting quantitative WMH volume assessment as a practical, scalable biomarker in clinical research and care [45,55,56].

We acknowledge several limitations of our study. First, our study had a relatively modest size of 41 participants, which may have limited statistical power, especially for detecting region-specific associations between WMH burden and cognitive performance. Second, our study was limited by the cross-sectional study design, which prevented us from investigating the effect of WMH volume on the progression of cognitive impairment. In the future, further investigation of results using longitudinal datasets would provide a better understanding of the significance of WMH. Third, WMH quantification relied on a fully automated segmentation software, which is objective and reproducible; however, it is dependent on predefined anatomical parcellation schemes and specific algorithms that vary between various tools and software [57]. Therefore, WMH volume estimates and related measurements may differ depending on the segmentation software used. Nevertheless, automated volumetric approaches are increasingly validated and offer substantial advantages for standardisation and longitudinal assessment.

## 5. Conclusions

Automated measurement of WMH volume was inversely related to MoCA score and separated cognitive groups clearer than the Fazekas scale. In our study, we observed statistically significant differences only between total WMH volume and MoCA groups. Visual grading showed weaker associations with cognition and poorer group discrimination than total WMH volume, whereas PVWM and DWM volumes did not show significant correlations with cognitive performance across groups. Total WMH volume rose stepwise across combined Fazekas grade groups, confirming convergent validity. Overall, quantitative automated WMH volumetry is the more objective, sensitive, and monitorable assessment tool for evaluating this biomarker in cognitive impairment. Future work should validate this in larger longitudinal cohorts to evaluate WMH progression and its relationship with cognitive decline. Future research could incorporate domain-specific cognitive testing and voxel-wise or tract-based analyses to further expand on the association between WMH and cognitive impairment.

## Figures and Tables

**Figure 1 medicina-62-00060-f001:**
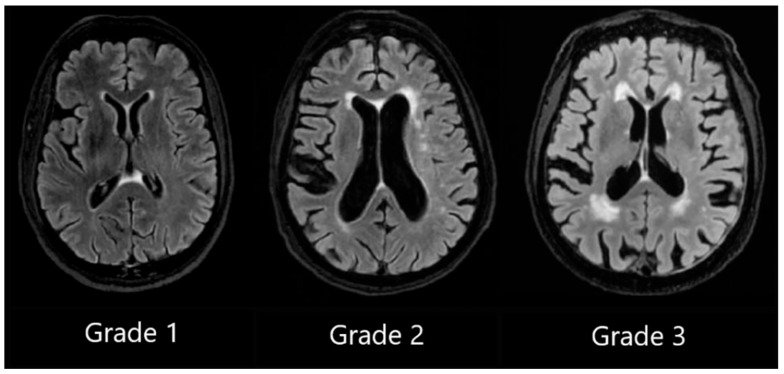
Periventricular white matter hyperintensities based on the Fazekas scale, ranging from Grade 1 to Grade 3, based on the severity of lesions.

**Figure 2 medicina-62-00060-f002:**
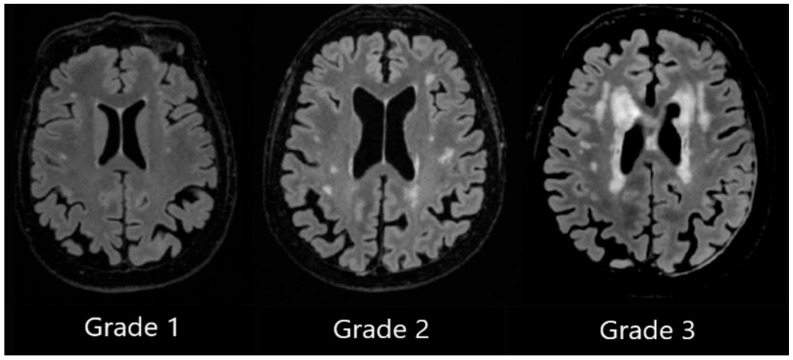
Deep white matter hyperintensities based on the Fazekas scale, ranging from Grade 1 to Grade 3, based on the severity of lesions.

**Figure 3 medicina-62-00060-f003:**
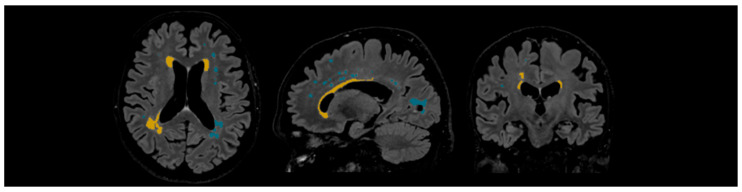
Example of automated lesion delineation in axial, sagittal, and coronal planes. Yellow colour—periventricular white matter hyperintensities; blue colour—subcortical white matter hyperintensities.

**Figure 4 medicina-62-00060-f004:**
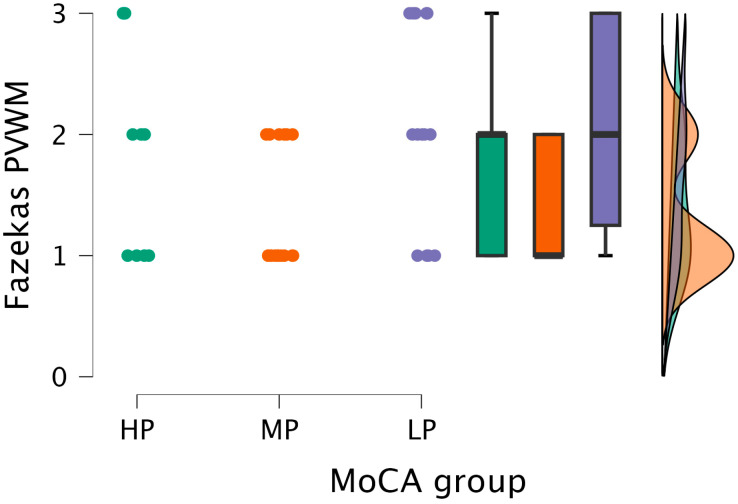
PVWM grade comparisons between (from left to right) HP (*n* = 9), MP (*n* = 18), and LP (*n* = 14) groups, with data distribution in each group.

**Figure 5 medicina-62-00060-f005:**
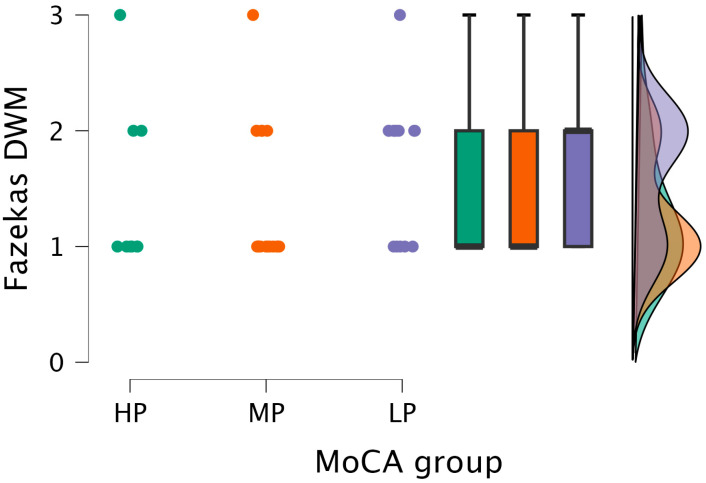
DWM grade comparisons between (from left to right) HP (*n* = 9), MP (*n* = 18), and LP (*n* = 14) groups, with data distribution in each group.

**Figure 6 medicina-62-00060-f006:**
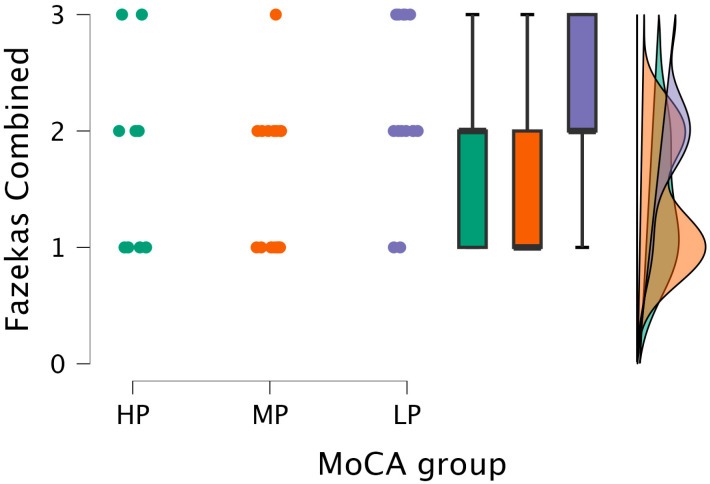
Combined grade (assigned based on the highest grade between PVWM and DWM lesions) comparisons between (from left to right) HP (*n* = 9), MP (*n* = 18), and LP (*n* = 14) groups, with data distribution in each group.

**Figure 7 medicina-62-00060-f007:**
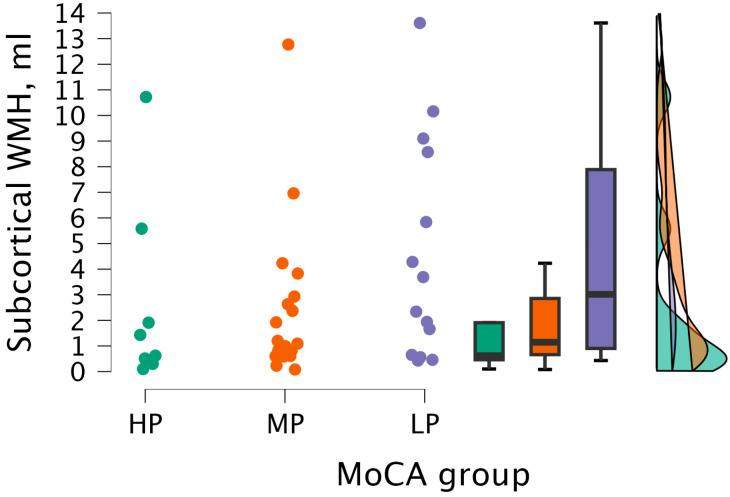
Subcortical WMH volume between (from left to right) HP (*n* = 9), MP (*n* = 18), and LP (*n* = 14) groups, with data distribution in each group.

**Figure 8 medicina-62-00060-f008:**
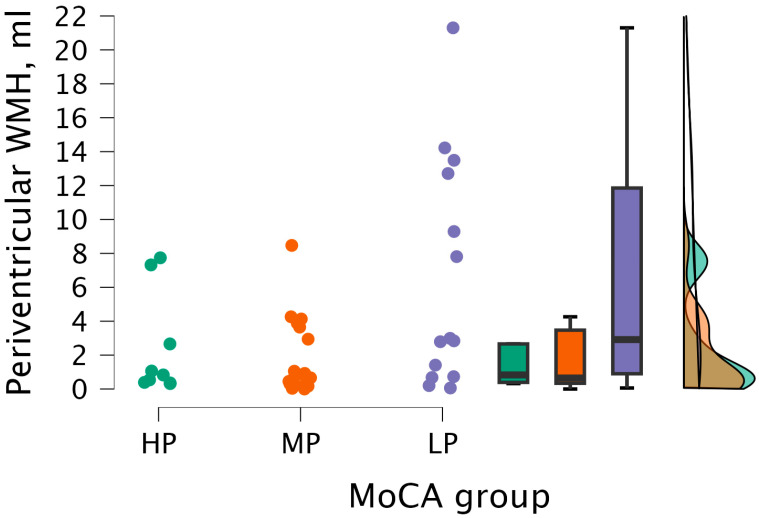
Periventricular WMH volume between (from left to right) HP (*n* = 9), MP (*n* = 18), and LP (*n* = 14) groups, with data distribution in each group.

**Figure 9 medicina-62-00060-f009:**
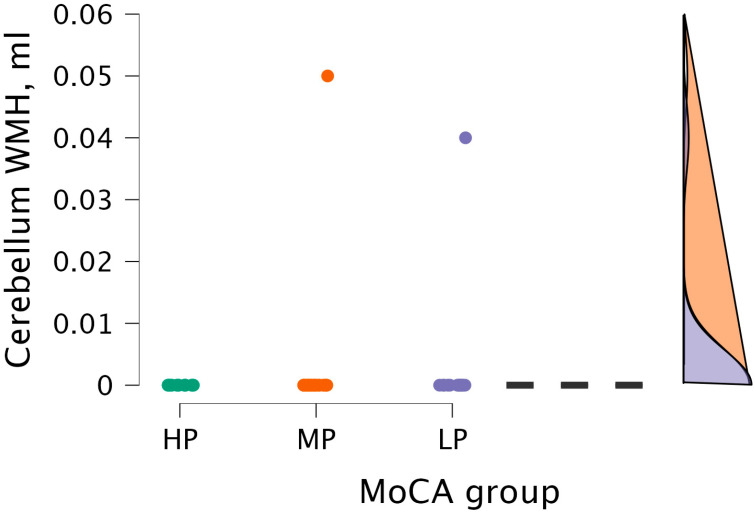
Cerebellum WMH volume between (from left to right) HP (*n* = 9), MP (*n* = 18), and LP (*n* = 14) groups, with data distribution in each group.

**Figure 10 medicina-62-00060-f010:**
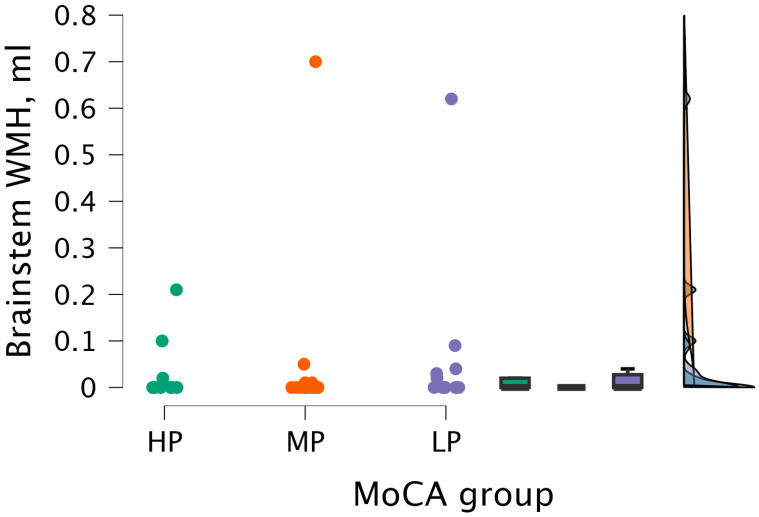
Brainstem WMH volume between (from left to right) HP (*n* = 9), MP (*n* = 18), and LP (*n* = 14) groups, with data distribution in each group.

**Figure 11 medicina-62-00060-f011:**
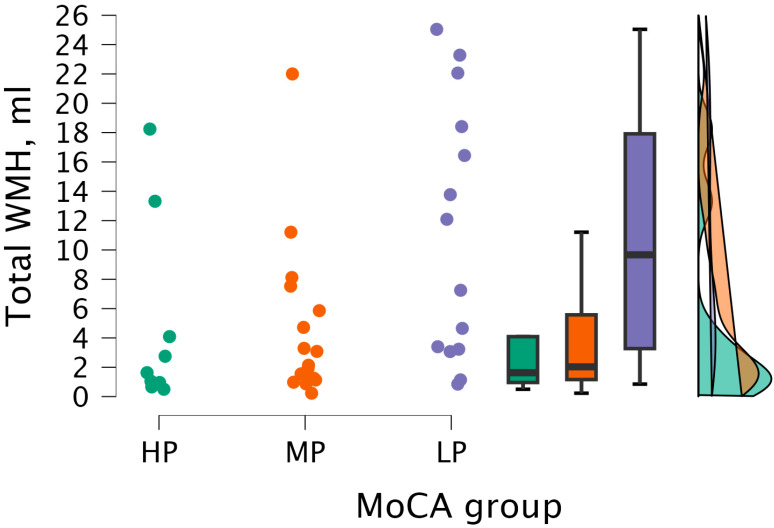
Total WMH volume between (from left to right) HP (*n* = 9), MP (*n* = 18), and LP (*n* = 14) groups, with data distribution in each group.

**Figure 12 medicina-62-00060-f012:**
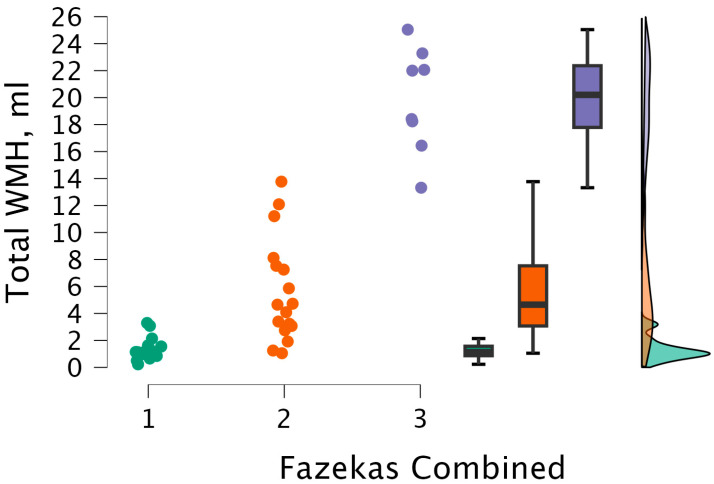
Total WMH volume between combined grade (assigned based on the highest grade between PVWM and DWM lesions) groups, with data distribution in each group. Combined grade 1 (*n* = 16); combined grade 2 (*n* = 17); combined grade 3 (*n* = 8).

**Table 1 medicina-62-00060-t001:** Results of tests for data normality for continuous variables.

	Age	MoCA	WMH Subcortical, mL	WMH Cerebellum, mL	WMH Periventricular, mL	WMH Brainstem, mL	Total, mL
N	41	41	41	41	41	41	41
Mean	71.049	19.488	3.161	0.002	3.525	0.046	6.735
Std. Deviation	10.312	8.000	3.681	0.010	4.837	0.146	7.442
Shapiro–Wilk	0.983	0.900	0.772	0.230	0.725	0.355	0.786
*p*-value of Shapiro–Wilk	0.804	0.002	<0.001	<0.001	<0.001	<0.001	<0.001
Minimum	44.000	4.000	0.080	0.000	0.000	0.000	0.230
Maximum	96.000	30.000	13.610	0.050	21.300	0.700	25.040

**Table 2 medicina-62-00060-t002:** Participant demographic data, gender, and MoCA results.

	Gender (F:M)			Age			MoCA		
	HP	MP	LP	HP	MP	LP	HP	MP	LP
N	8:1	11:7	9:5	9	18	14	9	18	14
Mean				65.0	70.3	75.9	28.2	22.7	9.7
Std. Error of Mean				3.8	1.7	3.0	0.4	0.5	1.2
Std. Deviation				11.5	7.4	11.1	1.1	2.3	4.4
Minimum				44	57	62	27	18	4
Maximum				77	81	96	30	25	16
χ^2^	2.296			4.702			34.802 ***		

*** = *p* < 0.001.

**Table 3 medicina-62-00060-t003:** Fazekas scale grade correlation with MoCA.

Spearman’s Correlations		
Variable		MoCA
1. MoCA	Spearman’s rho	—
	*p*-value	—
2. Fazekas PVWM	Spearman’s rho	−0.286
	*p*-value	0.070
3. Fazekas DWM	Spearman’s rho	−0.235
	*p*-value	0.139
4. Fazekas Combined	Spearman’s rho	−0.325 *
	*p*-value	0.038

* = *p* < 0.05.

**Table 4 medicina-62-00060-t004:** Dunn’s post hoc comparison of the PVWM grade and MoCA groups.

Comparison	z	W_i_	W_j_	r_rb_	*p*	p_bonf_	p_holm_
HP–MP	1.364	22.278	16.167	0.296	0.173	0.518	0.345
HP–LP	−0.877	22.278	26.393	0.198	0.380	1.000	0.380
MP–LP	−2.614	16.167	26.393	0.500	0.009 **	0.027 *	0.027 *

* = *p* < 0.05, ** = *p* < 0.01.

**Table 5 medicina-62-00060-t005:** Dunn’s post hoc comparison of the combined grade (assigned based on the highest grade between PVWM and DWM lesions) and MoCA groups.

Comparison	z	W_i_	W_j_	r_rb_	*p*	p_bonf_	p_holm_
HP–MP	0.862	20.444	16.528	0.179	0.388	1.000	0.388
HP–LP	−1.402	20.444	27.107	0.310	0.161	0.483	0.322
MP–LP	−2.669	16.528	27.107	0.524	0.008 **	0.023 *	0.023 *

* = *p* < 0.05, ** = *p* < 0.01.

**Table 6 medicina-62-00060-t006:** WMH volumes by region.

	WMH Subcortical, mL	WMH Periventricular, mL	WMH Cerebellum, mL	WMH Brainstem, mL	Total, mL
N	41	41	41	41	41
Mode	0.460	0.000	0.000	0.000	0.230
Median	1.660	1.050	0.000	0.000	3.230
Mean	3.161	3.525	0.002	0.046	6.735
Std. Error of Mean	0.575	0.755	0.002	0.023	1.162
Std. Deviation	3.681	4.837	0.010	0.146	7.442
IQR	3.630	3.680	0.000	0.010	10.070
Minimum	0.080	0.000	0.000	0.000	0.230
Maximum	13.610	21.300	0.050	0.700	25.040
25th percentile	0.600	0.450	0.000	0.000	1.140
50th percentile	1.660	1.050	0.000	0.000	3.230
75th percentile	4.230	4.130	0.000	0.010	11.210

**Table 7 medicina-62-00060-t007:** WMH volumes by region split by MoCA groups.

	MoCA Groups	N	Median	Mean	Std. Deviation	Minimum	Maximum
	HP	9	0.620	2.403	3.548	0.100	10.720
WMH subcortical, mL	MP	18	1.145	2.482	3.107	0.080	12.770
	LP	14	3.015	4.521	4.277	0.430	13.610
	HP	9	0.830	2.358	3.021	0.320	7.740
WMH periventricular, mL	MP	18	0.650	1.821	2.275	0.000	8.470
	LP	14	2.915	6.466	6.733	0.060	21.300
	HP	9	0.000	0.000	0.000	0.000	0.000
WMH cerebellum, mL	MP	18	0.000	0.003	0.012	0.000	0.050
	LP	14	0.000	0.003	0.011	0.000	0.040
	HP	9	0.000	0.037	0.073	0.000	0.210
WMH brainstem, mL	MP	18	0.000	0.043	0.164	0.000	0.700
	LP	14	0.000	0.057	0.164	0.000	0.620
	HP	9	1.630	4.800	6.446	0.500	18.240
Total, mL	MP	18	2.030	4.348	5.352	0.230	22.000
	LP	14	9.670	11.049	8.767	0.850	25.040

**Table 8 medicina-62-00060-t008:** Spearman’s correlation of lesion load and overall MoCA score.

Spearman’s Correlations by MoCA Scores		
Variable		MoCA
1. MoCA	Spearman’s rho	—
	*p*-value	—
2. WMH subcortical, mL	Spearman’s rho	−0.293
	*p*-value	0.063
3. WMH periventricular, mL	Spearman’s rho	−0.290
	*p*-value	0.066
4. WMH cerebellum, mL	Spearman’s rho	−0.066
	*p*-value	0.683
5. WMH brainstem, mL	Spearman’s rho	−0.030
	*p*-value	0.853
6. Total, mL	Spearman’s rho	−0.392 *
	*p*-value	0.011

* = *p* < 0.05.

**Table 9 medicina-62-00060-t009:** Dunn’s post hoc comparison of total WMH volumes and MoCA groups.

Comparison	z	W_i_	W_j_	r_rb_	*p*	p_bonf_	p_holm_
HP–MP	−0.420	16.111	18.167	0.123	0.674	1.000	0.674
HP–LP	−2.281	16.111	27.786	0.540	0.023 *	0.068	0.068
MP–LP	−2.253	18.167	27.786	0.484	0.024 *	0.073	0.068

* = *p* < 0.05.

**Table 10 medicina-62-00060-t010:** Dunn’s post hoc comparison of the total WMH volume between the combined grade (assigned based on the highest grade between PVWM and DWM lesions) groups.

Comparison	z	W_i_	W_j_	r_rb_	*p*	p_bonf_	p_holm_
1–2	−3.300	9.938	23.706	0.831	<0.001 ***	0.003 **	0.002 **
1–3	−5.290	9.938	37.375	1.000	<0.001 ***	<0.001 ***	<0.001 ***
2–3	−2.661	23.706	37.375	0.985	0.008 **	0.023 *	0.008 **

* = *p* < 0.05, ** = *p* < 0.01, *** = *p* < 0.001.

## Data Availability

The data that support the findings in this study are available from the corresponding author upon reasonable request.

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
