# Peer review of "Quantifying White Matter Hyperintensities: Automated Volumetry Compared with Visual Grading Scales"

_medicina, 2025, doi:10.3390/medicina62010060_

Round 1
Reviewer 1 Report
Comments and Suggestions for Authors
Comment 1
The study is well designed, but the Introduction would benefit from a clearer explanation of the specific clinical need addressed by comparing automated WMH quantification with the Fazekas scale. Highlighting the exact gap in the current literature would strengthen the rationale and better contextualize the study’s contribution.
Comment 2
Although the statistical approach is generally appropriate, the Methods section should provide additional details on how assumptions for the non-parametric tests were verified; whether outliers in WMH volumes were assessed or handled, and the criteria used to justify the selected tests.
Comment 3
Several analyses, particularly those involving regional WMH volumes (e.g., Figures 7–10), did not reach statistical significance. The Discussion should briefly address the clinical meaning of these non-significant results, including whether sample size or statistical power may have limited the ability to detect regional differences.
Comment 4
Some figures are visually strong, but others would benefit from improvements such as adding sample sizes to all graphs; standardizing axis scales across related figures, and providing clearer distribution markers.
Comment 5
The Discussion touches on relevant literature but should explore in more detail why only total WMH volume correlated with MoCA, while regional volumes did not show the same association. Offering possible mechanistic or methodological explanations would improve the depth and coherence of the Discussion section.
Author Response
Dear reviewer,
We extend our gratitude to you for taking the time to review our manuscript. We appreciate your insights, and please find the detailed responses to your comments below:
1) “The study is well designed, but the Introduction would benefit from a clearer explanation of the specific clinical need addressed by comparing automated WMH quantification with the Fazekas scale. Highlighting the exact gap in the current literature would strengthen the rationale and better contextualize the study’s contribution.”
Thank you for the feedback, we added an emphasis in the Introduction on why it is crucial to compare automated WMH quantification with the Fazekas scale and noted the current gap in available literature.
2) “Although the statistical approach is generally appropriate, the Methods section should provide additional details on how assumptions for the non-parametric tests were verified; whether outliers in WMH volumes were assessed or handled, and the criteria used to justify the selected tests.”
We included a table in the Materials and Methods section showing the results of the Shapiro-Wilk test. Because the data distribution was not normal, non-parametric tests were utilised to analyse the data further. WMH outliers were not removed; however, a comment in the statistical analysis subsection was added, encouraging readers to refer to confidence intervals of raincloud plots.
3) “Several analyses, particularly those involving regional WMH volumes (e.g., Figures 7–10), did not reach statistical significance. The Discussion should briefly address the clinical meaning of these non-significant results, including whether sample size or statistical power may have limited the ability to detect regional differences.”
We added a possible explanation of this finding. Thank you for the insight on this!
4) “Some figures are visually strong, but others would benefit from improvements such as adding sample sizes to all graphs; standardizing axis scales across related figures, and providing clearer distribution markers.”
Thank you for the comments! Sample sizes have been added to the graphs.
The axis scales were standardised where possible, but it is not feasible to standardise the regional WMH volume graphs due to significant differences in volume values. Therefore, a decision was made to use different maximum values for each graph to present them in the most visually understandable manner.
5) “The Discussion touches on relevant literature but should explore in more detail why only total WMH volume correlated with MoCA, while regional volumes did not show the same association. Offering possible mechanistic or methodological explanations would improve the depth and coherence of the Discussion section.”
We have added possible explanations of this observation based on your suggestion.
Thank you for taking the time to review this article!
Have a nice day!
Note – Changed or added text is highlighted in green. Text in grey was rephrased, as per the Editor’s request, and text highlighted in yellow remained unchanged.
Best regards,
Arturs Titovs

Reviewer 2 Report
Comments and Suggestions for Authors
8 December 2025
The review report on the manuscript, titled ‘Automated Quantitative Assessment and Anatomical Distribution of Cerebral White Matter Hyperintensities in Patients with Cognitive Impairment’by Titovs A, submitted to Medicina
Manuscript ID: medicina-4038136
Dear Authors,
White matter hyperintensities (WMHs) are commonly observed on brain MRI in individuals with cognitive impairment and are strongly linked to cerebral small vessel disease and neurodegeneration. Yet, clinical evaluation predominantly relies on visual grading methods like the Fazekas scale, which are limited by subjectivity and inter-rater variability, posing challenges for accurate, scalable, and reproducible assessment. In the current manuscript entitled ‘Automated Quantitative Assessment and Anatomical Distribution of Cerebral White Matter Hyperintensities in Patients with Cognitive Impairment,’ Titovs and colleagues investigate the relationship between the volume and anatomical distribution of white matter hyperintensities and cognitive impairment severity using automated quantitative MRI analysis.
A major strength of this study lies in its use of fully automated volumetric analysis to assess white matter hyperintensities. This method reduces subjectivity, ensures consistency, and enables precise regional measurements. Unlike traditional visual grading, it captures subtle differences in lesion burden. The approach enhances reliability, particularly in research settings requiring scalability. Altogether, it provides a more objective and reproducible link between brain imaging and cognitive performance.
This manuscript presents a timely and engaging study likely to appeal to Medicina’s readership. The topic is important and well framed, though the argument requires further depth and supporting evidence. Incorporating the recommended revisions would significantly improve the manuscript’s clarity and scholarly contribution.
Comments:
- Title: Kindly revise the title to make it concise, clear, and reflective of the study’s central message, as the title serves as one of the manuscript’s most critical elements. Suggested options include: Quantifying White Matter Hyperintensities: Automated Precision vs. Visual Grading Limitations in Cognitive Impairment; Mapping Mind and Matter: Automated Volumetry of White Matter Lesions in Cognitive Decline; From Subjective Scores to Scalable Metrics: Comparing Visual and Automated WMH Assessments in Dementia Risk.
- Abstract: To enhance clarity and coherence, the abstract should be revised with attention to structure, content balance, and brevity. Preferably, it must not exceed 200 words and should be written as one continuous paragraph without subheadings. Begin with a concise introduction to the broader topic, followed by a focused description of the specific context and the research gap the study addresses. Clearly articulate the study’s rationale and objectives to convey its scientific relevance. In describing methods and results, emphasize the core procedures and major findings, avoiding excessive detail. Conclude the results section by briefly situating the findings within a wider scientific context. The final portion of the abstract should begin with a strong, declarative sentence such as “Here we show,” summarizing the main outcome. This should be followed by a brief discussion of the study’s implications or potential applications. Close with two or three sentences that connect the findings to broader scientific questions or fields. Do not include statistical values.
- Keywords: Please include ten appropriate MeSH terms and integrate multiple keywords into both the title and the first two sentences of the abstract. This will enhance the manuscript's discoverability and ensure better alignment with indexing requirements.
- I highly recommend presenting an informative graphical or video abstract.
- Introduction: Refine this section to improve clarity and maintain engagement. Open with a broad conceptual overview that positions the study within its larger scientific context, then gradually narrow the focus to the specific problem being addressed. Identify research gaps that justify the investigation and lead logically to the study’s aims. Organize the content into well-structured paragraphs totaling approximately one thousand words, allowing space to fully develop core ideas. The writing should remain clear and accessible to readers across disciplines, while precisely defining the study’s purpose. Conclude the introduction with a succinct statement that underscores the significance of the work and sets up a smooth transition into the following sections of the manuscript.
- Methods: Open this section with a concise paragraph outlining the study’s design and methodological framework. To improve rigor, incorporate references that support key methodological choices. Well-selected literature will enhance the credibility of the approach and strengthen the foundation of the presented evidence. To enhance methodological rigor, the authors could specify whether data extraction was independently verified by multiple reviewers. Including inter-rater reliability or detailing how discrepancies were resolved would improve transparency. This clarification strengthens reproducibility and ensures that retrospective data collection was conducted with consistency and minimal risk of bias. To strengthen methodological transparency, consider detailing the preprocessing steps used by the automated software. Specify image normalization techniques, lesion detection thresholds, and quality control procedures. Include information on intra- and inter-software reliability if available. A flowchart outlining each image processing phase would also enhance reproducibility and clarity for future studies.
- Results: Enhance the results by including effect sizes alongside p-values to convey the magnitude of group differences. Clearly label all statistical comparisons in figures and tables. Consider adding confidence intervals for key findings. These additions will offer a fuller picture of statistical significance, clinical relevance, and the robustness of observed trends. Conclude with a structured paragraph that synthesizes the findings and highlights their broader significance. Exclude statistical details from the main text and direct readers to the relevant tables for numerical data, ensuring clarity and maintaining the focus on interpretation and relevance.
- Discussion: Present the discussion as a continuous narrative and structure it into several coherent paragraphs totaling around fifteen hundred words. Open with a contextual introduction and close with a focused synthesis of the main findings. Develop the discussion to clarify the study’s aims, challenges faced, and advances required to overcome them. Emphasize the broader research landscape and highlight how this work deepens or extends current knowledge. Address the wider significance of the results, particularly their potential to inform future studies. Conclude with a balanced evaluation of both strengths and limitations, while considering possible clinical relevance. This approach will support a cohesive, reflective, and engaging discussion that aligns with academic standards..
- Conclusion: To clarify the manuscript’s central contribution, consider adding a focused paragraph of approximately 180 to 200 words. This should demonstrate the authors’ expertise and offer a thoughtful synthesis of key insights. Emphasize both theoretical value and practical relevance, while identifying gaps that warrant further investigation. Addressing overlooked conceptual or methodological aspects can guide future research. These additions will enrich the narrative and enhance the manuscript’s overall coherence and impact..
- References: Consider adding more references, since research of this scale typically includes over sixty citations to ensure sufficient context and to substantiate the findings more effectively.
The manuscript contains nive figures, nine tables, and 55 references. This study provides valuable insight into the relationship between white matter hyperintensity burden and cognitive function using automated MRI analysis. By directly comparing volumetric data with traditional visual grading, it highlights the advantages of objectivity and reproducibility. The method captures subtle anatomical differences often missed by conventional scoring. Its clinical relevance is notable, especially for early detection and longitudinal monitoring. With rising dementia prevalence, this approach offers a scalable, data-driven tool to support precision diagnostics in cognitive impairment research and care. I hope that after careful revision, the manuscript meets the journal’s high standards for publication. In addition, I anticipate the authors preparing “a detailed point-point rebuttal” to my remarks.
I declare no conflict of interest regarding this manuscript.
Best regards,
Reviewer
Author Response
Dear reviewer,
We extend our gratitude to you for taking the time to review our manuscript. We appreciate your insights, and please find the detailed responses to your comments below:
1) “Title: Kindly revise the title to make it concise, clear, and reflective of the study’s central message, as the title serves as one of the manuscript’s most critical elements. Suggested options include: Quantifying White Matter Hyperintensities: Automated Precision vs. Visual Grading Limitations in Cognitive Impairment; Mapping Mind and Matter: Automated Volumetry of White Matter Lesions in Cognitive Decline; From Subjective Scores to Scalable Metrics: Comparing Visual and Automated WMH Assessments in Dementia Risk.”
Thank you for the suggestion. We have changed the title to reflect our study’s focus better.
2) “Abstract: To enhance clarity and coherence, the abstract should be revised with attention to structure, content balance, and brevity. Preferably, it must not exceed 200 words and should be written as one continuous paragraph without subheadings. Begin with a concise introduction to the broader topic, followed by a focused description of the specific context and the research gap the study addresses. Clearly articulate the study’s rationale and objectives to convey its scientific relevance. In describing methods and results, emphasize the core procedures and major findings, avoiding excessive detail. Conclude the results section by briefly situating the findings within a wider scientific context. The final portion of the abstract should begin with a strong, declarative sentence such as “Here we show,” summarizing the main outcome. This should be followed by a brief discussion of the study’s implications or potential applications. Close with two or three sentences that connect the findings to broader scientific questions or fields. Do not include statistical values.”
We revised the abstract, removing statistical values to ensure it was clearer and more concise. We included subheadings in the abstract per the Medicina manuscript template; we believe this helps showcase the paper in a visually clear way.
3) “Keywords: Please include ten appropriate MeSH terms and integrate multiple keywords into both the title and the first two sentences of the abstract. This will enhance the manuscript's discoverability and ensure better alignment with indexing requirements.”
We have updated the list of keywords to include eleven terms; our title and first paragraph contain multiple keywords, underscoring the focus of this manuscript.
4) “I highly recommend presenting an informative graphical or video abstract.”
Based on your suggestion, we have made a graphical abstract for the manuscript, which is viewable in the new version of our paper.
5) “Introduction: Refine this section to improve clarity and maintain engagement. Open with a broad conceptual overview that positions the study within its larger scientific context, then gradually narrow the focus to the specific problem being addressed. Identify research gaps that justify the investigation and lead logically to the study’s aims. Organize the content into well-structured paragraphs totaling approximately one thousand words, allowing space to fully develop core ideas. The writing should remain clear and accessible to readers across disciplines, while precisely defining the study’s purpose. Conclude the introduction with a succinct statement that underscores the significance of the work and sets up a smooth transition into the following sections of the manuscript.”
We have added the clinical relevance of our study and identified current research gaps to better reflect its significance.
6) “Methods: Open this section with a concise paragraph outlining the study’s design and methodological framework. To improve rigor, incorporate references that support key methodological choices. Well-selected literature will enhance the credibility of the approach and strengthen the foundation of the presented evidence. To enhance methodological rigor, the authors could specify whether data extraction was independently verified by multiple reviewers. Including inter-rater reliability or detailing how discrepancies were resolved would improve transparency. This clarification strengthens reproducibility and ensures that retrospective data collection was conducted with consistency and minimal risk of bias. To strengthen methodological transparency, consider detailing the preprocessing steps used by the automated software. Specify image normalization techniques, lesion detection thresholds, and quality control procedures. Include information on intra- and inter-software reliability if available. A flowchart outlining each image processing phase would also enhance reproducibility and clarity for future studies.”
Thank you for the comments! Fazekas score was assigned by a single specialist rater; moreover, our study’s aim was not to validate inter-rater reliability, therefore, this part was not emphasised. References to specific literature have been incorporated, and the main steps have been described in our manuscript. There are publications that state variability depends on the software, for example, in a study done by Tabei et al. (https://pubmed.ncbi.nlm.nih.gov/35691185/). However, we would like to add that our goal was not to evaluate inter- and intra-software reliability.
7) “Results: Enhance the results by including effect sizes alongside p-values to convey the magnitude of group differences. Clearly label all statistical comparisons in figures and tables. Consider adding confidence intervals for key findings. These additions will offer a fuller picture of statistical significance, clinical relevance, and the robustness of observed trends. Conclude with a structured paragraph that synthesizes the findings and highlights their broader significance. Exclude statistical details from the main text and direct readers to the relevant tables for numerical data, ensuring clarity and maintaining the focus on interpretation and relevance.”
The effect sizes, where applicable, are shown by Z values in various tables. In addition, confidence intervals are shown beside every raincloud plot.
The significance of the main findings is discussed in the discussion section. We have decided to include statistical details to make the data easier to understand and more presentable. Moreover, the data are presented in this way so readers are not forced to look through every table to find significant results.
8) “Discussion: Present the discussion as a continuous narrative and structure it into several coherent paragraphs totaling around fifteen hundred words. Open with a contextual introduction and close with a focused synthesis of the main findings. Develop the discussion to clarify the study’s aims, challenges faced, and advances required to overcome them. Emphasize the broader research landscape and highlight how this work deepens or extends current knowledge. Address the wider significance of the results, particularly their potential to inform future studies. Conclude with a balanced evaluation of both strengths and limitations, while considering possible clinical relevance. This approach will support a cohesive, reflective, and engaging discussion that aligns with academic standards.”
Thank you for the suggestions! We have updated the discussion to focus on the main findings of our study and refined several key paragraphs by adding additional information.
9) “Conclusion: To clarify the manuscript’s central contribution, consider adding a focused paragraph of approximately 180 to 200 words. This should demonstrate the authors’ expertise and offer a thoughtful synthesis of key insights. Emphasize both theoretical value and practical relevance, while identifying gaps that warrant further investigation. Addressing overlooked conceptual or methodological aspects can guide future research. These additions will enrich the narrative and enhance the manuscript’s overall coherence and impact.”
We have revised the conclusion based on your suggestions.
10) “References: Consider adding more references, since research of this scale typically includes over sixty citations to ensure sufficient context and to substantiate the findings more effectively.”
We have added more references while improving various parts of the manuscript.
Thank you for taking the time to review this article!
Have a nice day!
Note – Changed or added text is highlighted in green. Text in grey was rephrased, as per the Editor’s request, and text highlighted in yellow remained unchanged.
Best regards,
Arturs Titovs

Round 2
Reviewer 1 Report
Comments and Suggestions for Authors
Manuscript is ready for publication.
Reviewer 2 Report
Comments and Suggestions for Authors
17 December 2025
The 2nd review report on the manuscript, titled ‘Automated Quantitative Assessment and Anatomical Distribution of Cerebral White Matter Hyperintensities in Patients with Cognitive Impairment’by Titovs A, submitted to Medicina
Manuscript ID: medicina-4038136
Dear Authors,
I appreciate the authors’ comprehensive and thoughtful responses to the issues raised during the previous round of review. The revised manuscript is now well structured and clearly written, and it rigorously investigates the association between the volume and anatomical distribution of white matter hyperintensities and the severity of cognitive impairment using automated quantitative MRI analysis. The study meets the high scientific and editorial standards of the journal, and I look forward to future contributions from this research group.
Thank you for the opportunity to review this work.I declare no conflict of interest regarding this manuscript.
Best regards,
Reviewer